# Combined Transcriptome and Metabolome Analysis Reveals Adaptive Defense Responses to DON Induction in Potato

**DOI:** 10.3390/ijms24098054

**Published:** 2023-04-29

**Authors:** Biao Zhao, Hang Yu, Dan Liu, Jiaqi Wang, Xu Feng, Fumeng He, Tianshuai Qi, Chong Du, Linlin Wang, Haifeng Wang, Fenglan Li

**Affiliations:** 1College of Life Sciences, Northeast Agricultural University, Harbin 150030, China; zhaob35@outlook.com (B.Z.);; 2State Key Laboratory for Conservation and Utilization of Subtropical Agro-Bioresources, College of Agriculture, Guangxi University, Nanning 530005, China

**Keywords:** *Phytophthora infestans*, potato, DON, metabolome, transcriptome

## Abstract

*Phytophthora infestans* poses a serious threat to potato production, storage, and processing. Understanding plant immunity triggered by fungal elicitors is important for the effective control of plant diseases. However, the role of the potato stress response to *Fusarium* toxin deoxynivalenol (DON)-induced stress is still not fully understood. In this study, the metabolites of DON-treated potato tubers were studied for four time intervals using UPLC-MS/MS. We identified 676 metabolites, and differential accumulation metabolite analysis showed that alkaloids, phenolic acids, and flavonoids were the major differential metabolites that directly determined defense response. Transcriptome data showed that differentially expressed genes (DEGs) were significantly enriched in phenylpropane and flavonoid metabolic pathways. Weighted gene co-expression network analysis (WGCNA) identified many hub genes, some of which modulate plant immune responses. This study is important for understanding the metabolic changes, transcriptional regulation, and physiological responses of active and signaling substances during DON induction, and it will help to design defense strategies against *Phytophthora infestans* in potato.

## 1. Introduction

To date, more than 10,000 varieties of potato (*Solanum tuberosum* L.) have been cultivated in more than 160 countries and regions, making it the world’s most important non-cereal food crop [1,2]. Its tubers are rich in starch, dietary fiber, protein, minerals, vitamins, and other nutrients required by the human body [3]. As a high-yielding crop, potatoes are widely used as food ingredients, feed processing, bioenergy reserves, and in environmental material development, playing an important role in ensuring food security and increasing income. Global losses due to *Fusarium* infection account for 17.2% of total potato production, causing huge economic losses [4]. At moderate temperatures (16–22 °C) and high humidity (over 97%), the pathogen usually infects tubers by being washed down through soil [5]. Potatoes are susceptible to infection during planting and storage, resulting in red, brown or granular rot of the tubers, which destroys the entire crop within days of infection, severely affecting potato quality and yield. Storage rot can also occur if the pathogen is present, and infections can lead to secondary infections of fungi and bacteria. Although chemical control can play a better role in a short period of time, there are problems with resistance and pesticide residues. Biological control has the advantages of being environmentally friendly and less prone to resistance during disease prevention and control, and it is an important complementary means of chemical control. For example, tuber biocontrol of potato late blight was achieved using phenazine-1-carboxylic acid produced by *Pseudomonas* spp. [6]. Studies have shown that *P. infestans* also produces virulence factors (cell wall-degrading enzymes, virulence-related effector proteins, and phytotoxins) such as T-2 toxin, DON, ENN-A, ZEB, and FB, which can promote infestation while causing disease [7]. DON mainly contaminates cereals (maize, wheat, and barley etc.) and its stable structural properties have led to its detection in feed, cereal products, and by-products [8]. As one of the most dangerous natural food contaminants, DON inhibits protein biosynthesis, mainly by binding to the ribosomal peptidyl transferase [9]. Simultaneously, certain virulence factors can act as elicitors to trigger specific reactions and activate plant immunity, which has great application in plant disease control [10,11]. Studies have shown that host defence responses can be induced using low concentrations of DON, which has an elicitor-like effect [12,13].

Pathogens and elicitors can activate plant-induced defenses and produce heritable resistance [14,15]. In the process of plant resistance to pathogenic bacterial infestation, there is synthesis and secretion of secondary metabolites through the activation of non-specific defense response PTI and specific defense response ETI, which, in turn, can interact with each other to produce a more powerful plant defense response against pathogen infection, mainly in terms of related gene expression, altered enzyme activity, triggered signaling pathways, altered ion channels, NO and AOS and ROS bursts, and plant antitoxin accumulation [16,17]. For example, AsES can trigger defense responses against *P. griseofulvin* in *Arabidopsis* [18]; *Pseudomonas coronafaciens* secretes coronatine, which is structurally similar to natural hormones and can trigger specific gene expression and synthesis of alkaloids and isoflavonoid phytoantotoxins [19]; cryptogein secreted by *Phytophthora citrophthora* can induce a local hypersensitive response (HR) and systemic acquired resistance (SAR) in tobacco [20]; and the *Valsa mali* exciton, VmE02, induces tobacco cell death, ROS accumulation, callus deposition, and activation of salicylic acid (SA) and jasmonic acid (JA) induced immune responses [21].

DON, a virulence factor of *Fusarium*, contributes to the colonization of *P. infestans*, and the study of DON resistance can provide new insights into plant disease resistance [22]. Studies have shown that DON treatment in wheat activates the wheat methionyl tRNA synthetase (*TaMetRS*) gene, which encodes a novel MetRS with defense and detoxification effects [23]. *TaUGT6* enhances resistance to *Fusarium* head blight (FHB) and accumulation of DON in wheat [24]. Additionally, *TaCYP72A* enhances wheat resistance to DON [25]. DON acts as a precursor toxin to produce deoxynivalenol-3-glucoside (D3G) via D3G glycosylation reactions to exercise detoxification. The development of FHB and *Fusarium* crown rot (FCR) has been reported to be suppressed by limiting the action of DON during plant glycosylation [26,27].

When plants activate their defenses, secondary metabolites synthesized through primary metabolites or other biosynthetic pathways regulate adaptive plant growth, mainly including terpenoid, phenylpropanoid, alkaloid, and polyketide metabolic pathways [28]. The phenylpropane pathway produces various phenolic acids, terpenoids, and flavonoids with broad-spectrum antimicrobial activity, which play a key role in broad-spectrum disease resistance [29,30]. For example, the activity of phenylalanine aminolytic enzymes, hydroxycinnamic acid CoA reductase, and hydroxycinnamic acid dehydrogenase was enhanced after fungal infection of lily, and phenylalanine aminolytic enzyme (*PAL*) genes were found to be important genes for cell wall-mediated immunity and were involved in broad-spectrum disease resistance [31]. Activation of defense signals leads to the accumulation of SA, which, in turn, activates key plant disease resistance signals [32]. Different defense modes include the deposition of different glycans, callose [33], phenolics, and antimicrobial substances; defense against papilla formation [34]; lignification [35] and production of reactive oxygen species; and deposition of plant antitoxins against pathogenic bacteria [36]. For example, flavonoid biosynthesis plays an important role in the response of *Stylosanthes* to *Colletotrichum* infection [37]. Accordingly, pathogens can also use effector proteins to modulate plant immunity; for example, the Hrp-secretory apparatus of fire blight can precisely block the expression of enzyme genes encoding the phenanthrene metabolic pathway in susceptible genotypes of apple [38].

In recent years, combined transcriptional and metabolic analyses have become an important tool for discovering new signaling pathways and responses to environmental stress induction, and they are an effective method for revealing plant stress responses, plant-pathogen interactions, mechanisms of metabolite synthesis, and identification of resistance genes [39]. For example, transcriptome analysis of *P. infestans* infected potatoes revealed SA-JA-ABA(abscisic acid)-mediated signaling pathways [40]; Li et al. [41] elucidated the molecular mechanisms of metabolite accumulation and flavonoid biosynthesis in jujube leaves by metabolomic and transcriptomic analyses; and Yang et al. [42] elucidated the biosynthetic regulatory network of flavonoid metabolites in *Salvia miltiorrhiza* stems and leaves through transcriptional metabolic analysis.

Our previous study found that treatment of potato tubers with low concentrations of DON also affected resistance to late blight caused by *P. infestans* [43]. However, the effects of fungal toxins on potato tuber resistance and the relationship between resistance and transcriptional and metabolic changes have not been reported, and the molecular basis and physiological and biochemical changes in potato-induced resistance under DON induction are not clear. Therefore, in this study, RNA-seq technology and non-targeted metabolomics were used to study the expression patterns of genes and overall changes in metabolites under DON stress using Atlantic potato tuber as material. Four time points were taken under DON treatment at 5 ng/mL to reveal the specific genes regulating the resistance response under different treatment times as well as the transcriptional network and major metabolic pathways in response to DON stress in order to further understand three things––the mechanism of DON stress response in potato, the DEGs in the transcriptional network, and major metabolic pathways in response to DON stress.

## 2. Results

### 2.1. UPLC-MS/MS-Based Quantitative Metabolomic Analysis of the Atlantic Potato Cultivar Tuber

To better understand the dynamics of relevant metabolites during DON stress, total metabolites were extracted from each sample, and metabolite detection was performed using UPLC-MS/MS. The mass spectrometry data of each group of samples (CK0h, DON0h, DON4h, DON12h, and DON48h) were processed using Analyst 1.6.3 software, and a total of 676 metabolites were detected based on the local metabolic database. The integration of the peaks and the correction of the chromatographic peaks were performed on the downstream mass spectrometry results of each sample using MultiaQuant software (v3.0.3) to obtain the relative content of each substance. The 15 samples were clearly divided into 5 groups by heat map and correlation analysis (Figure 1A), and there was a significant correlation between the samples within the groups (Appendix A). Additionally, PCA showed clear separation between the groups (Figure 1B). As shown in Figure 1C, 676 metabolites could be divided into 13 different categories, but most were concentrated in lipids (17%), flavonoids (12%), alkaloids (12%), amino acids and derivatives (12%), organic acids (9%), and others. Overall, the results indicate that DON stress in potato has different metabolite distributions at different times.

### 2.2. Metabolomic Changes at Different Time Points

To show the changes in the metabolite content of potato tubers under different DON stress durations, we used orthogonal partial least squares discriminant analysis (OPLS-DA) to screen for differential metabolites, and metabolites with fold change ≥2 or ≤0.5, along with VIP ≥ 1, were considered as significantly differential metabolites. Compared with CK0h, 75, 117, 180, and 197 differential metabolites were identified in DON0h, DON4h, DON12h, and DON48h, respectively, and more upregulated metabolites were specific at 48 h (Figure 2). Among them, six metabolites were upregulated and 69 metabolites were downregulated in the DON0h treatment compared with the control. In DON4h, 75 and 42 metabolites were upregulated and downregulated, respectively. A total of 102 and 78 metabolites were upregulated and downregulated, respectively, in DON12h. In addition, 144 upregulated metabolites and 55 downregulated metabolites were identified in DON48h.

A total of 301 differentially accumulated metabolites (DAMs) in the different samples were classified into 12 categories, of which the main ones were flavonoids (18.3%), phenolic acids (16.6%), and alkaloids (16.2%). Most of the DAMs showed large differences at different times, with 55 of the 82 flavonoids showing variability and significant down-regulation at 0 h and 12 h; 50 of the 99 phenolic acid metabolites showed variability and significant accumulation at 12 h and 48 h; and 49 of the 83 alkaloids showed variability and significant accumulation at 48h (Figure 3A). Notably, there were 15 metabolites that were differentially synthesized at different times, suggesting that these 15 substances may play an important role in the response of potato to DON stress (Figure 3B), namely, indole (C00463), *N*-benzylformamide (C15561), calystegine A3 (C10850), 2-(formylamino) benzoic acid (C05653), l-tyrosine (C00082), 3-indoleacrylic acid (C21283), l-tryptophan (C00078), methoxyindoleacetic acid (C05660), *N*-acetyl-l-tryptophan, *N*-caffeoylputrescine (C03002), p-coumaroylmalic acid, 5-*O*-caffeoylshikimic acid (C10434), syringaldehyde-4-*O*-glucoside, laempferol-3-*O*-(2-*O*-xylosyl-6-*O*-rhamnosyl)glucoside, and quercetin-3-*O*-rutinoside. To further understand the in vivo interactions of the differential metabolites in plants, we annotated and enriched the functions of the differential metabolites based on the KEGG database and found that all the differential metabolites were mainly annotated in metabolic pathways and biosynthesis of secondary metabolites. The enrichment results showed that nine pathways—metabolic pathways, biosynthesis of secondary metabolites, ABC transporters, biosynthesis of amino acids, 2-oxocarboxylic acid metabolism, aminoacyl-tRNA biosynthesis, phenylpropanoid biosynthesis, arginine and proline metabolism, and cysteine and methionine metabolism—were enriched at various time periods under DON stress (Figure 3C).

Taken together, our results indicate that potatoes under DON stress mainly exhibit phenolic acid and alkaloid metabolite accumulation patterns, with significant differences in defense responses for approximately 50% of the metabolites that are directly or indirectly attributed to the phenylpropane metabolic pathway.

### 2.3. Transcriptional Analysis of Potato Tubers under DON Stress

To study the gene expression profile of potato tubers under DON stress at different times, we performed RNA-seq on the same samples as those used in metabolomics. Fifteen samples from five groups of treatments were sequenced with libraries, and each library of Clean Data was greater than 6 GB, Q30 base percentage was greater than 90%, average length was 150 bp, and GC content of each sample was greater than 43% (see Appendix A for raw sequencing data composition statistics and Appendix A for quality control details), indicating that the transcriptome sequencing results were reliable and suitable for further analysis. We mapped each sample (at least 47 million clean reads) using hisat2 software (v2.2.1), in which more than 95.85% of the reads were compared to the reference genome, with a Unique mapped ratio between 51.78% and 52.81% (Appendix A), and a Unique mapped ratio of approximately 50% may be due to the reference genome species being tetraploid resulted. Unique-mapped assemblies were quantified using the StringTie software (v2.10.5) to determine the count of each gene, which was reported as transcripts per kilobase per million (TPM, transcripts per kilobase of transcripts per million mapped reads). After normalization of expression, a total of 98,701 genes in at least one treatment with TPM > 0 were considered to be expressing genes, and were further analyzed (Appendix A).

To assess the reproducibility of the RNA-seq data, we calculated Pearson correlation coefficients between the three biological replicates for each sample. The results showed that the Pearson correlation coefficients between the three biological replicates ranged from 0.88 to 0.99, with large differences between groups, and PCA graph also showed good sample consistency (Appendix A), indicating that the transcriptomic data had a high reproducibility.

We identified DEGs between the control (CK0h) and other treatments using |log2 (fold change)| ≥ 1 and Padj < 0.05 as screening criteria. Compared with the control, 1053, 20,943, 20,347, and 21,172 were identified at 0 h, 4 h, 12 h, and 48 h under DON stress, respectively (Figure 4A). We found that most DEGs were identified under DON48h treatment, followed by DON4h, DON12h, and DON0h, compared with CK0h. The fewest DEGs were identified at DON0h, and the number of upregulated genes was twice as many as the number of down-regulated genes. The other three treatments (DON48h, DON12h, and DON0h) had approximately 20 times more DEGs than at DON0h. The number of upregulated DEGs and the number of downregulated DEGs were similar, indicating that DON induced significant changes in potato tuber gene expression after 4 h (Figure 4B).

Analysis of DEGs at different treatment times showed that each treatment time had its specific DEGs, and the highest number of specific DEGs was found at DON4h, followed by DON48h treatment, whereas 97 and 30 consistently upregulated and downregulated, respectively, and expressed DEGs were found in the four sets of comparisons, indicating that these genes may play a crucial role in response to DON stress in potato (Figure 4C,D). Expression pattern analysis showed that most DEGs under DON stress showed specific expression over time.

Gene Ontology (GO) and KEGG enrichment analyses of DEGs were performed. All protein sequences were annotated based on the EGGNOG website, and the Orgdb package was constructed to functionally annotate and enrich each group of DEGs according to GO classification. In the GO analysis of the CK0h-DON0h group, 420, 328, and 391 DEGs were classified as biological processes, molecular functions, and cellular components, respectively. In the GO analysis of the CK0h-DON12h group, 7854, 6783, and 7846 DEGs were classified into biological processes, molecular functions, and cellular components, respectively. In the GO analysis of the CK0h-DON48h group, 8013, 6928, and 7991 DEGs were classified as biological processes, molecular functions, and cellular components, respectively. Compared with the CK0h control, DEGs under DON stress from 0 to 48 h were significantly enriched in the secondary metabolite biosynthetic process (GO:0044550), phenylpropanoid biosynthetic process (GO:0009699), and phenylpropanoid metabolic process (GO:0009698) (Figure 5, Appendix A).

DEGs were annotated in the KEGG database to understand their biological functions and gene interactions. In the CK0h-DON0h group, 423 of the 1053 DEGs were enriched in 27 pathways, including phenylpropanoid biosynthesis, flavonoid biosynthesis, and phenylalanine metabolism (Appendix A). In the CK0h-DON4h group, 5495 of the 20,943 DEGs were enriched in 79 pathways, including phenylpropanoid biosynthesis, flavonoid biosynthesis, and photosynthesis antenna proteins (Appendix A). In the CK0h-DON12h group, 5928 of the 20,943 DEGs were enriched in 91 pathways, including photosynthesis-antenna proteins, phenylpropanoid biosynthesis, and photosynthesis (Appendix A). In the CK0h-DON48h group, 6036 of the 21,172 DEGs were enriched in 100 pathways, including phenylpropanoid biosynthesis, photosynthesis antenna proteins, and flavonoid biosynthesis (Appendix A). Notably, phenylpropanoid and flavonoid biosynthesis consistently occupied an important position in the enrichment of DEGs in KEGG during 0 to 48 h of DON stress, which is consistent with GO and metabolomic analyses. Therefore, phenylpropanoid and flavonoid biosynthesis may be the main pathways in response to DON stress.

### 2.4. Weighted Gene Co-Expression Network Analysis (WGCNA)

Genes with very low expression (average TPM < 1) were removed from this analysis to avoid the inclusion of spurious edges in the networks. We screened 52,387 highly expressed DEGs from 63,883 DEGs using co-expression network analysis. The power values were first filtered to ensure that the gene distribution conformed to a scale-free network. Hierarchical clustering of all the samples showed no outlier samples. When the power value (β value) was 20, several genes with similar expression profiles belonging to the same subnetwork were clustered into the same co-expression modules (Figure 6A and Appendix A). Finally, 14 modules were generated, with module sizes ranging from 60 to 13,318 genes, of which 1461 genes were not clustered into co-expression modules (Figure 6B). In addition, 400 genes were selected for visualization to determine the correlation within the module genes (Appendix A).

We investigated the correlation between differential metabolites and 14 co-expression modules and found that three modules were significantly correlated with 15 substances (|r| > 0.80, *p* < 0.01): darkseagreen1 (6 positive correlations), darkseagreen3 (3 positive correlations), and darkgoldenrod3 (1 positive correlation), and seven modules showed a clear trend of positive expression. Among all modules, the darkseagreen1 module was significantly correlated with seven differential metabolites, suggesting that genes in this module are highly correlated with the resistance mechanism of potato plants under DON stress. KEGG enrichment of three highly correlated modules revealed that the phenylpropanoid biosynthesis, oxidative phosphorylation, cysteine and methionine metabolism, necroptosis, and phenylalanine metabolism pathways were significantly enriched (Figure 6C), suggesting that these pathways may play an important role in the stress response of potato to DON.

A scale-free network is only linked to a few nodes whose connectivity can represent the importance of the nodes. Therefore, we used the MCC algorithm in Cytoscape 3.9.1 software cytohubba plug-in to identify all hub genes from the three highly correlated modules separately and took the top 20 genes as hub genes (Figure 6D–F, Appendix A).

### 2.5. Validation of Single-Gene Expression

To further verify the accuracy of the data analysis, we randomly selected 15 of these 60 hub genes for qRT-PCR validation. The results showed that the expression of each gene obtained by qRT-PCR analysis correlated with the TPM values obtained from the RNA-seq data, confirming the reliability of the RNA-seq data (Figure 7).

## 3. Discussion

Late blight caused by *P. infestans* occurs in nearly all potato-growing regions and is a major devastating disease for potato production of which the extent of damage is determined by the local climate of the year [44]. Prevention is made more difficult by the rapid mutation of the pathogenic fungus to adapt to the environment in which it survives, by the development of resistance, or by the resulting weakened resistance of resistant potato varieties [45]. Plant activation of PTI by conserved PAMP molecules recognized by PRRs and induction of ETI by pathogen effectors recognized by resistance proteins (R proteins) can enhance plant resistance and adaptation to biotic or abiotic stresses, which involve genes and pathways associated with different mechanisms [46]. Recently, a high-quality genome of the Atlantic variety of potato has been reported, which may facilitate our understanding of the role of stress-induced resistance in potato [47]. In our laboratory, we preliminarily found that low concentrations of DON could induce resistance to *P. infestans* in potatoes. To understand the metabolic changes and regulatory mechanisms of active compounds during DON induction, we constructed transcriptomic and metabolomic profiles of potato tubers at different time points under low concentration treatment combined with broad target metabolomic profiles to investigate DEGs and differentially accumulated metabolites to explore the mechanisms associated with DON-induced resistance in potatoes.

Studies have shown that DON and nivalenol (NIV) convert toxins to their glucosides (DON3G and NIV3G) via xenobiotic detoxification [48]. DON treatment induces ROS production in wheat. DON treatment rapidly induced the transcription of some defense genes in a concentration-dependent manner [49,50]. Additionally, it causes H_2_O_2_ production in the grain, which triggers antimicrobial defense but may also stimulate cell death. Cloning DON resistance genes or transgenic detoxification-related genes can improve wheat blast resistance [51]. In 5 ng/mL DON treatment, we found that DEGs at all four time points were enriched in plant stress-related pathways, such as phenylpropane metabolism and synthesis and secondary metabolite synthesis, suggesting that the mechanism of potato defense induced by DON stress is mainly in the phenylpropane pathway. Multiple KEGG pathways were significantly enriched, with phenylpropanoid biosynthesis being the most abundant pathway, followed by flavonoid biosynthesis. These changes in the transcriptome correlated with metabolomic data. However, DON is still more toxic, and the number of virulence factors is a key element in inducing disease resistance in plants using excitons. Some studies have shown that both 3-keto-DON and 3-epi-DON show lower toxicity in converting DON to less toxic derivatives, and exhibit similar gene expression profiles to DON under 3-keto-DON treatment, which provides crucial information for future transgenic plant applications and development [52].

JA and SA are the major signaling pathways in plants, and these two signaling pathways are usually antagonistic; however, synergistic or additive effects have also been reported [53,54]. In this study, key genes associated with JA transduction, such as *JAR1* and *JAZ*, were upregulated, while *COI1* and *MYC2* were significantly downregulated, and key genes associated with SA transduction, such as *NPR1*, *TGA*, and *PR-1*, were significantly upregulated (Appendix A). Substance accumulation was also reflected in the metabolic data. As reported in existing studies, DON induction can enhance plant defense by regulating the gene expression of SA and JA pathways, causing SA accumulation and activation of pathogenesis-related genes (PR) [55]. Therefore, we hypothesize that DON induces plant defense through the interaction of SA and JA signaling, and that this process is regulated by DNA methylation [43,56].

Many changes occur in the primary metabolisms of plants, such as amino acids, nucleotides, and their derivatives, in response to stress, and the composition and levels of amino acids affect the resistance of plants [57,58]. Many plant defense compounds are derived from amino acid precursors, such as secondary metabolites and thioglucosides [59]. In the present study, 31 amino acids and their derivatives were significantly altered in potato tubers following DON induction. Among them, upregulation was mostly seen in toxic amino acids or sulfur-containing amino acids, such as l-tyramine, *S*-(5′-adenosyl)-l-homocysteine, *S*-adenosyl-l-methionine, and l-alanyl-l-phenylalanine, which may play a key role in DON-induced plant defense. Consistent with this finding, genes related to these amino acids and their derivatives were also induced.

Plant secondary metabolites such as alkaloids, phenolic compounds, and flavonoids are typical plant antitoxins that play an important role in protection against diseases [60]. However, the content and distribution of secondary metabolites in different species differ greatly [61]. In our metabolomic analysis, the metabolites varied at different times, with changes in phenolic acids, flavonoids, terpenoids, lignans, and coumarins. Notably, phenolic acids, alkaloids, and flavonoids accounted for the highest percentage of the differential metabolites in each comparison group. Similarly, Isayenka et al. [62] found that the phytotoxin thaxtomin A (TA) induced browning of potato tuber flesh due to the accumulation of phenolic compounds, which may play a role in protecting potato tubers from *Streptomyces scabiei*. In addition, it is possible that because the family of tyrosine-derived metabolites is flavonoids, DON induction caused significant downregulation of l-tyrosine, resulting in a reduction of 84% of all flavonoids in DAMs, suggesting that flavonoid levels may be negatively correlated with DON-induced plant defense. Combined with the transcriptome enrichment results, the phenylpropane synthesis pathway is a rich source of plant metabolites and a starting point for the production of many other important compounds such as alkaloids, coumarins, and lignans [31]. Therefore, the accumulation of gene expression and metabolites associated with phenylpropanoid biosynthesis suggests that the defense mechanisms after DON induction may be related to phenylpropanoid metabolism. This is consistent with the findings of Yogendra et al. [63,64] that metabolic pathways associated with the biosynthesis of phenylpropanoids, flavonoids, and alkaloids were strongly induced.

WGCNA divided all DEGs into 14 modules, and three modules were highly correlated with 15 shared DAMs. KEGG enrichment analysis showed that genes of related modules were mainly enriched in phenanthrene biosynthesis, which is consistent with the above results. The annotation of hub genes was mainly for functional protein-encoding genes such as *ATTPS6* and *COMT* [65]. Most hub genes in the three modules were related to the regulation of plant immune responses. For example, *COMT* in the darkseagreen3 module is a key enzyme in phenylpropanoid biosynthesis, catalyzing a variety of phenylpropanoids and flavonoids, including caffeic acid, caffeol, and 5-hydroxyferulic acid, and catalyzes the production of melatonin from *N*-acetyl-5-hydroxytryptamine [66]. The *ASMT* encoded *N*-acetylserotonin methyltransferase in the darkseagreen1 module is the final enzyme in the biosynthetic pathway that produces melatonin. *ASMT* is induced in rice by both ABA and methyl jasmonic acid treatment [67,68]. Studies have shown that melatonin administration ameliorates the toxic effects of DON in mice [69]. *MYB48* encoded transcription factor plays an important role in regulating stress response in response to JA signaling [70], and a previous study showed that *MYB48* plays a central role in improving tomato tolerance [71] and a positive response of *MYB48* to stress treatment in rice [72,73]. The cytochrome P450 family is not only involved in various metabolic pathways, such as alkaloids, terpenoids, and phenylpropanoids to form resistance substances in plants and improve plant resistance, but also acts as an antitoxin detoxification enzyme in plants [74]. *CYP71B36* and *CYP76C2* belong to the cytochrome P450 family, of which *CYP71B* has been shown to be involved in terpene biosynthesis (nerol and geraniol). For example, *CYP71BJ1* in rose, which forms part of the pathway leading to 19-*O*-acetyl horhammericine, will help elucidate how this branch point is controlled [75]. *CYP76C* is a sesquiterpene plant antitoxin hydroxylase that hydroxylates rishitin in potato to form rishitin-M1 [76]. *ABCG22* encodes an ABC transporter protein and the ABC transporter protein *TaABCC3.1* has been reported to contribute to DON tolerance [77]. qRT-PCR assays showed altered expression of these hub genes at different times under DON treatment, suggesting that these hub genes may play an important role in DON-induced plant immune responses. This study has important implications for the design of defense strategies against *P. infestans* in potatoes.

## 4. Materials and Methods

### 4.1. Plant Materials and Treatments

In this study, Atlantic variety potato tubers were used as experimental material (provided by Heilongjiang Academy of Agricultural Sciences, Harbin, China). To study the dynamic response of the whole potato genome and metabolome to DON stress, potato tubers were treated with 5 ng/mL Vomitoxin (DON, purchased from FERMENTEK, Jerusalem Israel) and sampled at 0 h, 4 h, 12 h, and 48 h (recorded as DON0h, DON4h, DON12h, DON48h), and 0 h under sterile water treatment as control (recorded as CK0h), and all samples were frozen in liquid nitrogen and stored at −80 °C. Three biological replicates were performed for all five groups of samples.

### 4.2. Metabolite Profiling Using UPLC-MS/MS

The preserved samples were freeze-dried under vacuum using a lyophilizer (Scientz-100F, SCIENTZ Biotechnology Co., Ningbo, China) and the freeze-dried samples were crushed to powder form using a grinder (MM 400, Retsch, Verder Shanghai Instruments and Equipment Co., Ltd., Shanghai, China) at 30 Hz for 90 s. 100 mg of sample powder was weighed and dissolved in 1.2 mL of 70% methanol extract at 4 °C, and vortexed every 30 min for 30 s for a total of 6 times, and the samples were placed in a refrigerator at 4 °C overnight. After centrifugation at 12,000 rpm for 10 min, the supernatant was aspirated, and the sample was filtered through a microporous membrane (0.22 μm pore size) and stored in the injection vial for subsequent UPLC-MS/MS analysis.

For each sample, three biological replicates were analyzed independently. Chromatography mass spectrometry acquisition was performed using ultra performance liquid chromatography (UPLC, SHIMADZU Nexera X2, Kyoto, Japan) and tandem mass spectrometry (MS/MS, Applied Biosystems 4500 QTRAP, Foster, CA, USA).LIT and triple quadrupole (QQQ) scans were performed by a triple quadrupole linear ion trap mass spectrometer (Q TRAP), AB4500 Q TRAP UPLC/MS/MS system. The chromatographic column was an Agilent SB-C18 (1.8 µm, 2.1 mm × 100 mm); mobile phase was ultra-pure water in phase A (with 0.1% formic acid): acetonitrile in phase B (with 0.1% formic acid); and elution gradient: 5% in phase B at 0.00 min, linearly increasing to 95% in phase B at 9.00 min and maintaining at 95% for 1 min (10.00–11.10 min). The B-phase ratio decreased to 5% and equilibrated at 5% until 14 min, with a flow rate 0.35 mL/min, a column temperature of 40 °C, and injection volume 4 μL. For the mass spectrometry analysis [78], the ESI source operating parameters were as follows: ion source, turbo spray; source temperature, 550 °C; ion spray voltage (IS), 5500 V (positive ion mode)/−4500 V (negative ion mode); ion source gas I (GSI), gas II (GSII), and curtain gas (CUR) were set to 50, 60 and 25.0 psi, respectively, and the collision-induced ionization parameters were set to high. The instrument was tuned and mass calibrated in QQQ and LIT modes with 10 and 100 μmol/L polypropylene glycol solutions, respectively. QQQ scans were performed using MRM mode with the collision gas (nitrogen) set to medium. Further DP and CE optimization was accomplished for individual MRM ion pairs, and a specific set of MRM ion pairs was monitored in each period based on the metabolites eluted within each period.

Metabolites were characterized and quantified based on Metware Biotechnology Ltd.’s own database MWDB and public metabolite databases (MassBank, KNAPSAcK, HMDB, MoToDB, ChemBank, PubChem, NIST Chemistry Webbook and METLIN) [79]. The MS data were characterized by a precise comparison of precursor ion (Q1), product ion (Q3) values, and retention time (RT). The metabolites were quantified by MRM analysis using QQQ mass spectrometry. The characteristic ions of each substance were screened by triple quadruple rods, and the signal intensities (CPS) of the characteristic ions were obtained in the detector. The integration and calibration of the peaks were performed on the sample off-board mass spectrometry files using MultiaQuant software (v3.0.3), the peak area (Area) of each peak represented the relative content of the corresponding substance, and the integrated data of the area of all peaks were obtained [80].

### 4.3. Metabolome Data Analysis

Principal component analysis for unsupervised pattern recognition was performed on the filtered data using the prcomp function (parameter scale = TRUE) in R software (v4.2.1) [81]; orthogonal partial least squares discriminant analysis (OPLS-DA) was performed by the OPLSR.Anal function in MetaboAnalyst 4.0 software, based on the OPLS-DA results, combined with the difference fold change values; and the VIP values of OPLS-DA model were used to screen out the differential metabolites based on the OPLS-DA results, and the screening criteria were: metabolites with fold change ≥2 and fold change ≤0.5 and VIP ≥ [82]. Metabolites with significant differences were normalized by unit variance scaling (UV), and heatmaps were drawn by the pheatmap package. The differential metabolites were annotated and enriched using the KEGG database [83].

### 4.4. RNA-Seq Analysis

Total RNA was extracted from each sample, and RNA-seq libraries were constructed using Illumina Stranded Total RNA Prep (three biological replicates per treatment) based on the Illumina Novoseq 6000 system, and the fragment size and concentration of the libraries were detected by Agilent 2100 Bioanalyzer. QC of the RNA-seq raw data was performed using FastQC (v0.11.9, https://www.bioinformatics.babraham.ac.uk/projects/fastqc/, accessed on 27 October 2022) and Trimmomatic (v0.36) to obtain clean reads [84], which were mapped to the Atlantic potato reference genome [47] by Hisat2 (v2.2.1) [85]. Expression level quantification was performed using FeatureCounts (v2.10.5) and expression normalization was performed with edgeR software (v3.36.1) using the TPM (transcripts per kilobase per million mapped reads) algorithm [86,87]. Differential gene identification was performed using the default parameters of the DESeq2 (v1.38.0) package in R (v4.2.1), and DEG identification was based on a cutoff value of |log2(fold change)| ≥ 1 as a screening criterion [88]. GO annotation was performed through the EGNOG online website (eggNOG-mapper (embl.de) for functional annotation of all protein sequences in the Atlantic potato genome [89], the AnnotationForge package built the protein annotation result files into the OrgDB package, and the self-built package was used to perform GO ORA enrichment analysis of DEGs. The enricher package was used to perform KEGG enrichment analysis on the DEGs. All results were visualized using R.

### 4.5. Weighted Gene Co-Expression Network Analysis (WGCNA)

Gene co-expression networks were constructed using R software version 4.2.1 with WGCNA (v1.71) [90]. Lowly expressed genes (TPM < 1) in DEGs were removed and co-expression networks were constructed by WGCNA analysis. Based on Pearson correlation coefficient calculation, the signed function was transformed to construct the correlation matrix between genes (threshold 0.8). The correlation matrix was transformed into an adjacency matrix by the power adjacency function. To characterize the nonlinear relationship between genes, a topological overlap matrix (TOM matrix) was constructed from TOM correlation coefficients, through R software package blockwise Modules function, to build the coexpression network by dividing modules and merging similar modules to the module feature vector (MEs matrix) and character matrix in order to analyze the correlation between the module feature genes and DON processing different time/control processing, which would estimate the correlation of the module character [91]. The hub genes were calculated by Cytoscape software (v3.7.2) and network visualization was completed [92].

### 4.6. Real-Time Quantitative PCR (qRT-PCR) Validation

Fifteen genes were randomly selected from hub genes for qRT-PCR analysis using the LineGene 9620 Real-Time PCR system from BIOER, with β-action as an internal reference gene. These genes include *ABCG22*, *CYP71B36*, *SR1IP1*, *C3H20*, *PG2*, *CKX1*, *CYP76C2*, *VSP3*, *APD2*, *ADAP*, *COMT*, *RBF1*, *PTPMT1*, *WD40*, *L15e*, and the primers are shown in Appendix A. Total RNA was extracted from each treated sample using the Total RNA Rapid Extraction Kit (ER501-01, TransGen Biotech, Beijing, China) and reverse transcribed to cDNA using the TransScript^®^ One-Step gDNA Removal and cDNA Synthesis SuperMix (AT311, TransGen Biotech, Beijing, China). qRT- PCR reactions were performed using the ChamQ Universal SYBR qPCR Master Mix (Q711, Vazyme, Nanjing, China) kit. data from three independent replicates were calculated by the 2^−ΔΔCt^ method [93].

## Figures and Tables

**Figure 1 ijms-24-08054-f001:**
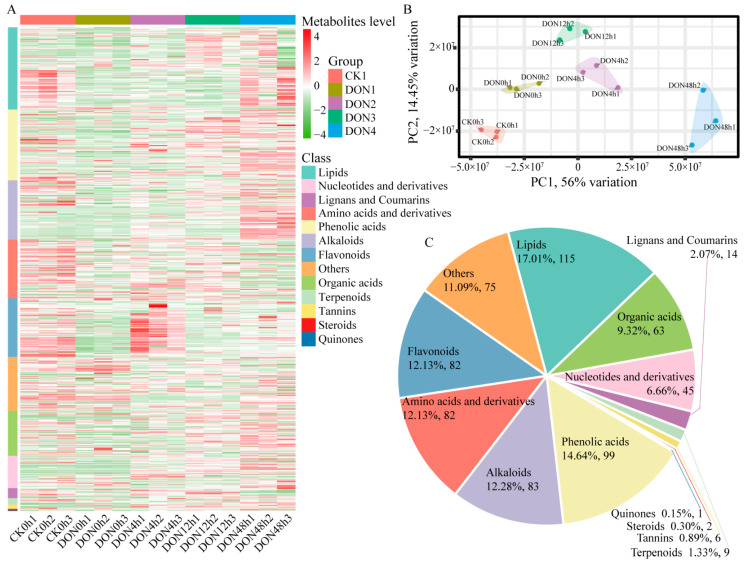
Qualitative and quantitative analysis of metabolomics data from samples. (**A**) Heatmap of metabolites for quantitative identification. The color scale indicates the level of metabolite accumulation and class indicates the metabolite class. (**B**) Principal component analysis. (**C**) Component analysis of metabolites.

**Figure 2 ijms-24-08054-f002:**
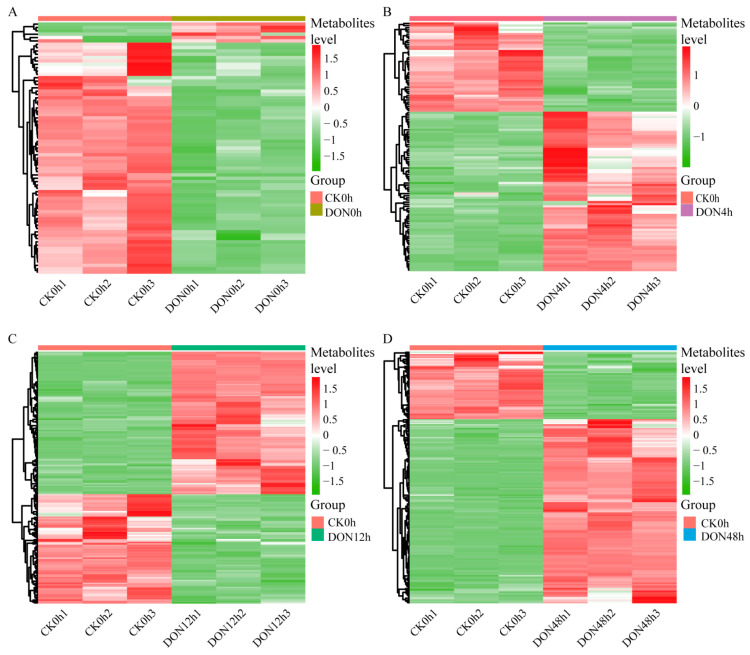
Differential accumulation of metabolites between CK0h vs. DON0h at 4, 12, and 48 h, with color scales indicating the level of metabolite accumulation. (**A**) Heatmaps of DAMs for CK0h vs. DON0h. (**B**) Heatmaps of DAMs for CK0h vs. DON4h. (**C**) Heatmaps of DAMs for CK0h vs. DON0h. (**D**) Heatmaps of DAMs for CK0h vs. DON0h.

**Figure 3 ijms-24-08054-f003:**
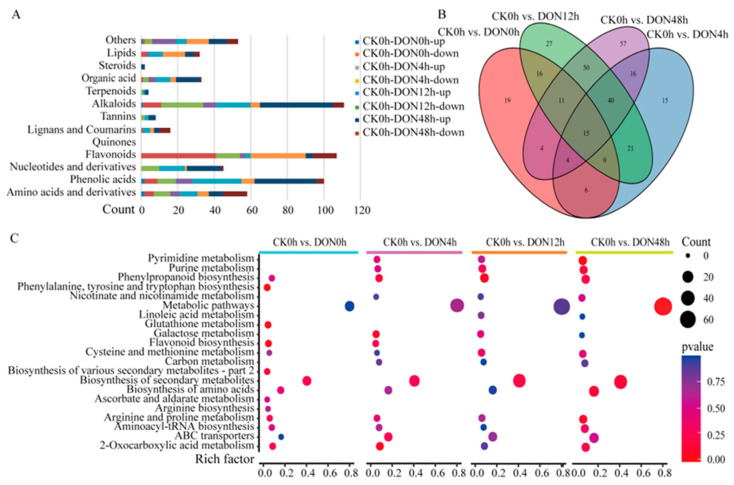
Accumulation of DAMs in different samples. (**A**) Histogram of the classification of DAMs in different samples. (**B**) Venn diagram showing the number of differential metabolites in each group. (**C**) KEGG enrichment plot of differential metabolites.

**Figure 4 ijms-24-08054-f004:**
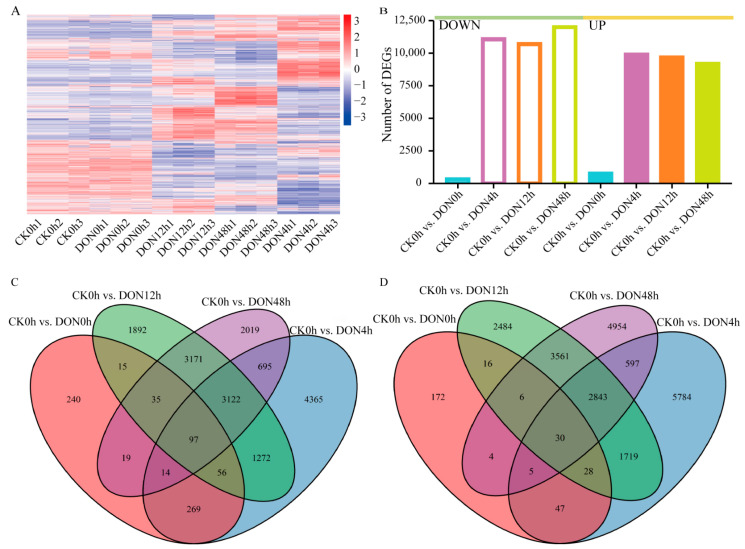
DEGs at different time points after DON treatment. (**A**) Heatmaps representing hierarchically clustered DEGs. (**B**) Statistical plot of DEGs. (**C**,**D**) Venn diagram of the number of DEGs, with C indicating Up and D indicating Down.

**Figure 5 ijms-24-08054-f005:**
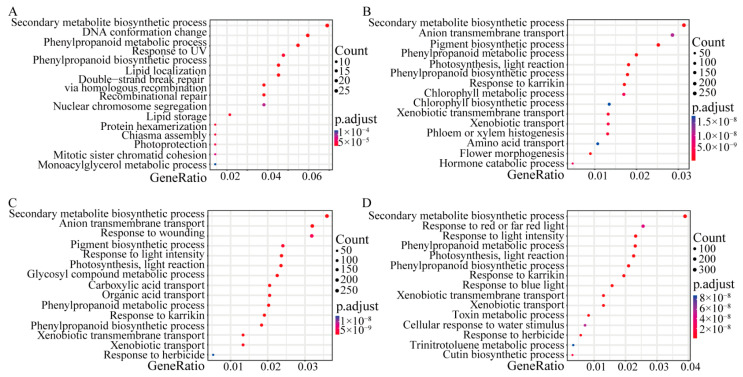
GO functional enrichment map of DEGs under DON stress. (**A**) GO Functional enrichment map of DEGs in the CK0h vs. DON0h groups. (**B**) Functional enrichment map of DEGs GO in the CK0h vs. DON4h groups. (**C**) Functional enrichment map of DEGs GO in the CK0h vs. DON12h groups. (**D**) Functional enrichment map of DEGs GO in the CK0h vs. DON48h groups.

**Figure 6 ijms-24-08054-f006:**
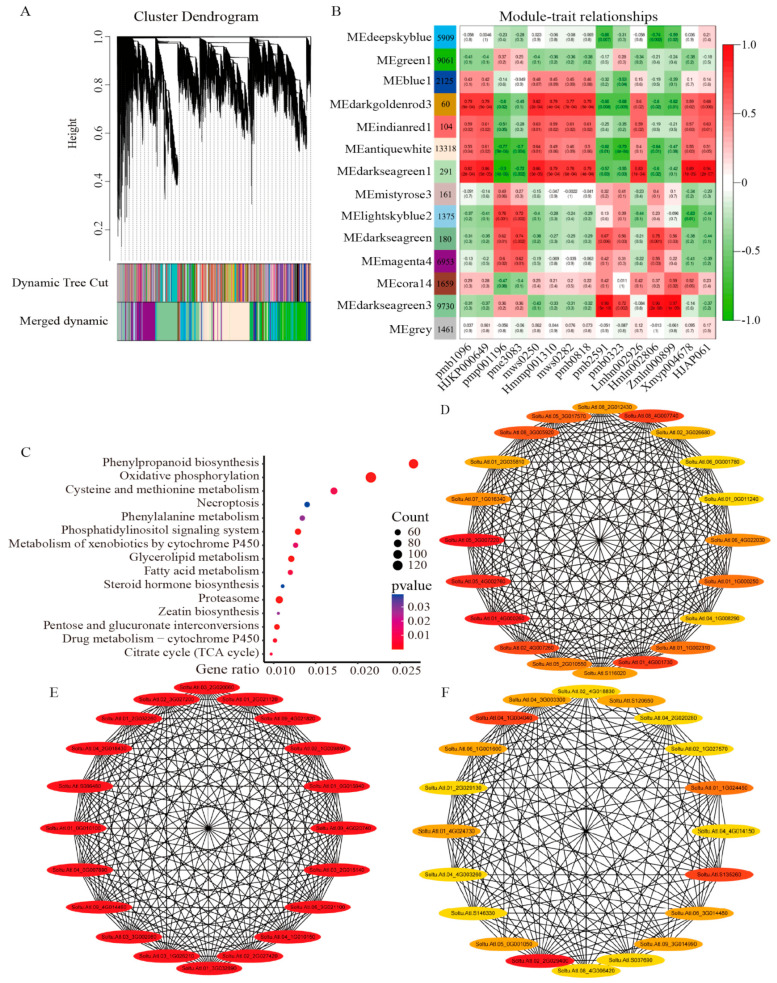
WGCNA of DEGs. (**A**) Hierarchical clustering tree with 14 modules. A leaf in the tree represents the DEGs, and each major branch is represented by a module. Note that the module MEgrey indicates an unassigned gene. (**B**) Correlation between modules-metabolites and corresponding *p*-values (in parentheses). The left color panel shows the number of individual modules and module member genes. The right color scale shows module-metabolite correlations. (**C**) Bubble plot of KEGG enrichment analysis of metabolic pathways of darkseagreen1, darkseagreen3, and darkgoldenrod3 module members, showing the top fifteen pathways. (**D**–**F**) Co-expression network analysis of shared differential metabolite-related modules, with the hub genes being the first 20 nodes of the (**D**) darkseagreen1, (**E**) darkseagreen3, and (**F**) darkgoldenrod3 modules.

**Figure 7 ijms-24-08054-f007:**
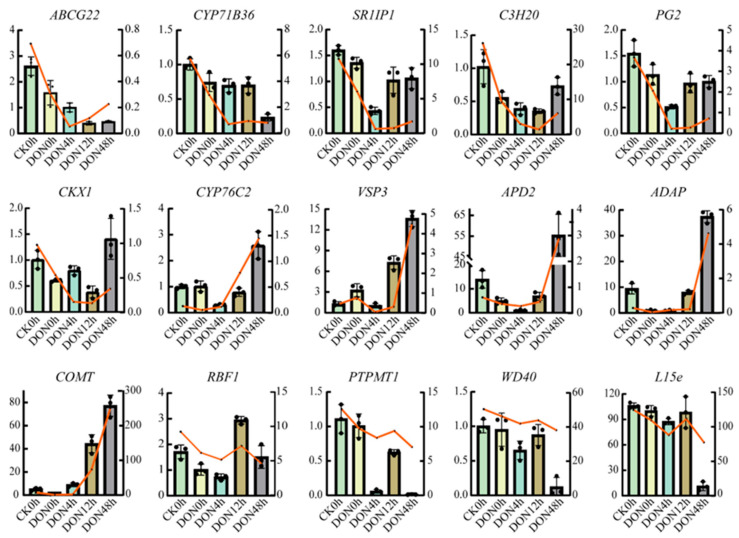
Expression profiles of genes detected by RNA-seq and qRT-PCR. The X-left axis represents qRT-PCR results (relative quantity calculated by 2^−ΔΔCT^) and Y−right axis represents RNA-seq results (TPM). The line chart shows the gene expression levels from the transcriptome (TPM) and the bar chart shows the gene expression from qRT-PCR.

## Data Availability

The RNA-seq datasets in this study are available from the NCBI Sequence Read Archive under project PRJNA943451. Metabolomics sequencing data are shown in Appendix A.

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
