# Peer review of "Combined Transcriptome and Metabolome Analysis Reveals Adaptive Defense Responses to DON Induction in Potato"

_ijms, 2023, doi:10.3390/ijms24098054_

Round 1

Reviewer 1 Report

How did the authors determine the low dose of DON?

Add negative impacts of DON on cereal production to the introduction.

Description of late blight symptoms in tubers needs improvement.  Symptoms result in reddish, brown, granular rot.  The pathogen usually infects tubers by being washed down through soil.  Storage rot can also occur if the pathogen  is present and infection can lead to secondary infections of fungi and bacteria.  The authors indicate 60% of  potato is infected - this value does not seem correct/appropriate to me.  If would also be helpful to add optimal environmental conditions for spread of this pathogen.

Reviewer 2 Report

The manuscript of Zhao et al. is a good study on a topical issue. Interesting data were obtained, which are quite well described and interpreted. An important remark that prevents me from recommending the manuscript for publication in its current form is the following.

The authors have identified an unprecedented large number of differentially expressed genes that need to be explained. Hoopes et al., cited by the authors (https://www.cell.com/molecular-plant/fulltext/S1674-2052(22)00003-X), analyzed allelic variants of potato tetraploids. 55,528 allelic groups were identified, which included about 80 percent of the genes of the studied variety. This is consistent with the fact that 61,422 genes have been found in the Atlantic core proteome. At the same time, the amount of DEGs determined by Zhao et al. in response to DON treatment is distributed over the samples in the range of 20.347 - 21.172, not counting the 0h point. Thus, about a third of all genes change their expression. At the same time, the authors found a change in the content of less than 200 metabolites in the studied samples.

Previously, the problems of constructing a variety-specific potato transcriptome were considered in detail (https://www.nature.com/articles/s41597-020-00581-4). Possible causes of transcriptome redundancy and ways to eliminate it were also considered. I recommend that you refer to this work and make changes to the data processing pipeline to avoid possible errors.

As specific steps, I would recommend the following:

1) First you need to add a link to the genome and the annotation that was used for mapping. Apparently, the genome from the Spud DB database was used.

http://spuddb.uga.edu/ATL_v3_download.shtml

https://www.ncbi.nlm.nih.gov/genome/annotation_euk/Solanum_tuberosum/100/

In this case, if the distribution of genes by alleles is not taken into account and a specific haploid variant is not chosen, then >20,000 genes in differential expression may well be associated with a large number of allelic isoforms. Since the construction of the cultivar-specific transcriptome has not yet been completed, this may be acceptable at this stage. However, it is necessary to indicate on which genome model the study was carried out and give its brief description.

2) 50% of uniquely matched reads don't seem very good. In fact, this percentage for potatoes should be around 70-80% (https://bmcplantbiol.biomedcentral.com/articles/10.1186/s12870-022-03461-8), so most likely there were some problems with sample preparation or mapping.

In general, Hisat2 is a pretty good alignment tool, but others may be useful for eukaryotes, such as https://www.ncbi.nlm.nih.gov/pmc/articles/PMC6275443/. For polyploids, it is better to use the STAR (https://github.com/alexdobin/STAR) or Stampy (https://www.well.ox.ac.uk/research/research-groups/lunter-group/lunter-group/stampy) mappers, especially if you are doing genome mapping. In addition, Kallisto (https://pachterlab.github.io/kallisto/) can be used to analyze differential expression, which focuses not on the genome, but on the transcriptome.

To control the consistency of the samples, in addition to the heat map, a PCA graph should be plotted.

In any case, it is necessary to explain the reason for the discrepancy between the presented dramatic changes in the transcriptome and the range of the plant's functional response.

Minor comment:

References should be made to works that have already studied the effect of DON on plants using transcriptomic and metabolomic approaches, for example:

https://www.frontiersin.org/articles/10.3389/fpls.2019.01137/full
